# Impaired Glycolysis Promotes Alcohol Exposure-Induced Apoptosis in HEI-OC1 Cells via Inhibition of EGFR Signaling

**DOI:** 10.3390/ijms21020476

**Published:** 2020-01-11

**Authors:** Hyunsook Kang, Seong Jun Choi, Kye Hoon Park, Chi-Kyou Lee, Jong-Seok Moon

**Affiliations:** 1Department of Otorhinoaryngology-Head and Neck Surgery, College of Medicine, Soonchunhyang University, Cheonan-si, Chungcheongnam-do 31151, Korea; ssook4311@hanmail.net (H.K.); akas9238@hanmail.net (S.J.C.); earpark@gmail.com (K.H.P.); 2Department of Integrated Biomedical Science, Soonchunhyang Institute of Medi-bio Science (SIMS), Soonchunhyang University, Cheonan-si, Chungcheongnam-do 31151, Korea

**Keywords:** glycolysis, EGFR signaling, apoptosis, alcohol, HEI-OC1 cells

## Abstract

Glucose metabolism is an important metabolic pathway in the auditory system. Chronic alcohol exposure can cause metabolic dysfunction in auditory cells during hearing loss. While alcohol exposure has been linked to hearing loss, the mechanism by which impaired glycolysis promotes cytotoxicity and cell death in auditory cells remains unclear. Here, we show that the inhibition of epidermal growth factor receptor (EGFR)-induced glycolysis is a critical mechanism for alcohol exposure-induced apoptosis in HEI-OC1 cells. The cytotoxicity via apoptosis was significantly increased by alcohol exposure in HEI-OC1 cells. The glycolytic activity and the levels of hexokinase 1 (HK1) were significantly suppressed by alcohol exposure in HEI-OC1 cells. Mechanistic studies showed that the levels of EGFR and AKT phosphorylation were reduced by alcohol exposure in HEI-OC1 cells. Notably, HK1 expression and glycolytic activity was suppressed by EGFR inhibition in HEI-OC1 cells. These results suggest that impaired glycolysis promotes alcohol exposure-induced apoptosis in HEI-OC1 cells via the inhibition of EGFR signaling.

## 1. Introduction

In cellular glucose metabolism, glycolysis is a critical pathway for energy production used for cell growth and proliferation [1,2]. Among the various enzymes in glycolysis pathway, hexokinase (HK) catalyzes the phosphorylation of glucose to produce glucose-6-phosphate (G-6-P) for the production of ATP and metabolites as the first rate-limiting step in glycolysis pathway [3,4]. In four HK isozymes (HK1–4), the expression of HK1 has been identified in various cell types [5,6,7,8]. The dysregulation of glucose metabolism has been linked to hearing loss [9,10].

Ligands and receptors from the epidermal growth factor (EGF) family were linked to cell proliferation in auditory cells [11,12]. EGF promotes hair cell differentiation from dissociated embryonic cochlear precursors [13] and promotes the proliferation of dissociated mouse cochlear supporting cells [14]. Additionally, other studies have implicated that the downstream signaling molecules of the epidermal growth factor receptor (EGFR) signaling pathway, such as phosphoinositide 3-kinase (PI3K)-AKT, could contribute to cell proliferation and survival [15,16,17]. 

Alcohol exposure causes damage to and functional impairment of various organs, including the liver, pancreas, cardiac muscle, and brain [18,19,20,21,22]. Alcohol metabolism-driven intermediates such as acetaldehyde and acetate contribute to cell death and tissue damages. Among alcohol metabolism-driven intermediates, acetaldehyde, a reactive and toxic metabolite of alcohol metabolism, is associated with the disturbance of physiological functions in various cells [23,24,25,26]. Recent studies demonstrated that alcohol can directly induce apoptotic pathways of neurons [27,28,29]. Alcohol exposure-induced neurotoxicity and neuronal degeneration were detected in a mouse model of ethanol exposure. Also, alcohol exposure contributes to the dysregulation of cellular energy metabolism, including glucose metabolism, in brain and liver [30,31,32]. A recent paper indicated that excessive alcohol consumption was linked to patients with hearing loss [33]. However, mechanisms by which the impaired glycolysis contributes to alcohol exposure-induced hearing loss remains incompletely understood.

In the current study, we demonstrate that the impaired glycolysis promotes alcohol exposure-induced apoptosis in HEI-OC1 cells, which is one of the few mouse auditory cells [34,35,36], via inhibition of EGFR signaling. We show that the inhibition of EGFR-induced glycolysis is a critical mechanism for alcohol exposure-induced apoptosis in HEI-OC1 cells. The cytotoxicity via apoptosis was significantly increased by alcohol exposure in HEI-OC1 cells. The glycolytic activity and the levels of HK1 were significantly suppressed by alcohol exposure in HEI-OC1 cells. Mechanistic studies showed that the levels of EGFR and AKT phosphorylation were reduced by alcohol exposure in HEI-OC1 cells. Notably, the HK1 levels and glycolytic activity were suppressed by EGFR inhibition in HEI-OC1 cells. These results suggest that impaired glycolysis promotes alcohol exposure-induced apoptosis in HEI-OC1 cells via inhibition of EGFR signaling.

## 2. Results

### 2.1. Alcohol Exposure Induced Cytotoxicity in HEI-OC1 Cells

To investigate whether the impaired glycolysis by alcohol exposure could promote cytotoxicity in HEI-OC1 cells, we first analyzed the morphological changes using 3D analyzer in HEI-OC1 cells. Moderate alcohol exposure, defined as the amount in two alcoholic beverages per day for men and one for women, equivalent to a blood alcohol content (BAC) of 0.04%, inhibited the function of intestinal epithelial cells [37]. Since it is not clear to determine the level of inner ear alcohol content by alcohol exposure in BAC of alcohol content, we used two concentrations of alcohol exposure with 0.01% (low) and 0.05% (high) in our system as low inner ear alcohol content and high inner ear alcohol content. Alcohol exposure increased the morphological features of cell death such as shrinkage, blebbing and losing contact with neighboring cells relative to control in dose-dependent manner (Figure 1A). Moreover, alcohol exposure significantly reduced the number of cells which were exhibited normal spindle shape in dose-dependent manner (Figure 1B). Next, we analyzed the changes of cell viability by alcohol exposure in HEI-OC1 cells. Cell cytotoxicity was measured by lactate dehydrogenase (LDH) released into the culture media from damaged or dead cells (Figure 1C). Notably, alcohol exposure significantly increased the levels of LDH relative to control in dose-dependent manner (Figure 1C). These results suggest that alcohol exposure induced cytotoxicity in HEI-OC1 cells.

### 2.2. Alcohol Exposure Induced Apoptosis in HEI-OC1 Cells

Next, we investigated whether alcohol exposure could induce a specific cell death pathway such as apoptosis or necrosis. We measured the number and proportion of dead or dying cells by double staining with Annexin V, which is used for detecting a loss of membrane asymmetry associated with early apoptosis, and 7-aminoactinomycin D (7AAD), which is used for the early phases of necrosis, in HEI-OC1 cells. Alcohol exposure significantly increased the apoptotic death cells (Annexin V-positive/7-AAD-negative) compared to control in dose-dependent manner (Figure 2A). The number of cells staining for apoptosis (Annexin V-positive/7-AAD-negative) was significantly increased from 5.4% in control to 10.1% (1.87 fold) after 0.01% alcohol treatment or 90.4% (16.7 fold) after 0.05% alcohol treatment (Figure 2B,C). In contrast, the number of cells staining for necrosis (Annexin V-negative/7-AAD-positive) was comparable between control and alcohol treatment (Figure 2B,C). Consistently, the levels of cleaved caspase-3, which is activated in the apoptotic cell, were significantly increased by alcohol exposure in dose-dependent manner (Figure 2D). These results suggest that alcohol exposure induced apoptosis in HEI-OC1 cells.

### 2.3. HK1-Dependent Glycolysis Is Suppressed by Alcohol Exposure in HEI-OC1 Cells

We next investigated the underlying molecular mechanism by which alcohol exposure regulates cytotoxicity via apoptosis in HEI-OC1 cells. Since the dysregulation of glucose metabolism has been linked to hearing loss [9,10], we examined whether the inhibition of glycolysis could be a critical metabolic change for alcohol exposure-induced apoptosis in HEI-OC1 cells. We first analyzed the changes of glycolytic activity by alcohol exposure in HEI-OC1 cells. We measured the extracellular acidification rate (ECAR) as the parameter of glycolysis activity by the quantification of lactate production [38,39]. The glycolytic activity was measured by the sequential addition of glucose, which is the substrate of glycolysis, oligomycin, a selective inhibitor for mitochondrial respiration, and 2-deoxyglucose (2-DG), a specific inhibitor of glycolysis [38,39]. We measured the change of ECAR levels by alcohol exposure of 0.01% and 0.05% in HEI-OC1 cells (Figure 3A,B). The levels of ECAR in response to glucose were significantly suppressed by alcohol exposure in dose-dependent manner relative to control (Figure 3A,B). Next, we investigated that the targets in the regulation of glycolysis by alcohol exposure in HEI-OC1 cells. We analyzed the changes of HK1 protein levels, the first key enzyme in the glycolysis pathway, by alcohol exposure in HEI-OC1 cells. Consistent with the reduction of glycolytic activity, the HK1 protein levels were suppressed by alcohol exposure in dose-dependent manner compared to control (Figure 3C). Notably, the levels of HK1 protein were significantly suppressed by alcohol exposure of 0.01% and 0.05% relative to control (Figure 3C). These results suggest that HK1-dependent glycolysis is suppressed by alcohol exposure in HEI-OC1 cells.

### 2.4. The Levels of EGFR and AKT Phosphorylation Were Reduced by Alcohol Exposure in HEI-OC1 Cells

We investigated that the molecular mechanism in the regulation of glycolysis during alcohol exposure-induced apoptosis in HEI-OC1 cells. Since EGFR-dependent signaling pathway is critical for glucose metabolism [40,41], we examined whether the EGFR-dependent signaling pathway could regulate glycolysis during alcohol exposure-induced apoptosis in HEI-OC1 cells. We first analyzed the changes of EGFR protein levels by alcohol exposure in HEI-OC1 cells (Figure 4A). The EGFR protein levels were reduced by alcohol exposure compared to control (Figure 4A). Consistently, the phosphorylation of AKT, which is a critical for the activation of glycolysis as a downstream signaling molecule of EGFR, was suppressed by alcohol exposure relative to control (Figure 4B). These results suggest that the expression of EGFR and AKT phosphorylation was reduced by alcohol exposure in HEI-OC1 cells.

### 2.5. Inhibition of EGFR Suppressed HK1-Dependent Glycolysis in HEI-OC1 Cells

To investigate the role of EGFR signaling in the regulation of HK-dependent glycolysis, we inhibited the EGFR signaling in HEI-OC1 cells using by Erlotinib HCl, a potent EGFR inhibitor. We first analyzed whether Erlotinib HCl could suppress the levels of HK1 in HEI-OC1 cells. The treatment of Erlotinib HCl (5 μM) suppressed the levels of HK1 compared to control (Figure 5A). To determine the inhibitory effect of Erlotinib HCl on glycolysis in HEI-OC1 cells, we measured the levels of ECAR in HEI-OC1 cells treated with Erlotinib HCl. Consistent with the HK1 levels, Erlotinib HCl (5 μM) significantly inhibited the levels of ECAR in response to glucose compared to control (Figure 5B,C). Moreover, the low dose treatment of Erlotinib HCl (1 μM) also suppressed the levels of HK1 and ECAR compared to control (Appendix A). These results suggest that inhibition of EGFR suppressed HK1-dependent glycolysis in HEI-OC1 cells. In summary, our results demonstrate that alcohol exposure suppresses the glycolysis via EGFR-HK1 pathway in HEI-OC1 cells (Figure 5D).

## 3. Discussion

In our study, we demonstrate that impaired glycolysis promotes alcohol exposure-induced apoptosis in HEI-OC1 cells via inhibition of EGFR signaling. We show that the inhibition of EGFR-mediated glycolysis is a critical mechanism for alcohol exposure-induced apoptosis in HEI-OC1 cells. The HK1-dependent glycolysis was suppressed by alcohol exposure in HEI-OC1 cells. The levels of EGFR and AKT phosphorylation were reduced by alcohol exposure in HEI-OC1 cells. Furthermore, the levels of HK1 and glycolytic activity were suppressed by EGFR inhibition in HEI-OC1 cells. Our results suggest that the inhibition of EGFR-mediated glycolysis could be an important metabolic pathway for alcohol exposure-induced apoptosis in HEI-OC1 cells.

Since the normal functions of hair cells require high levels of glucose and ATP for the modulation of hearing [42], the impairment of glucose metabolism related to diabetes mellitus (DM) might be associated with patients with hearing loss [43,44]. Previous studies have shown that the relationship between diabetes and hearing dysfunction [43,44]. Their results found that patients with diabetes have a high prevalence of hearing loss [43]. Further, the reduction of distortion product otoacoustic emissions amplitudes (DPOAEs), as a reflection of outer hair cell integrity and cochlear function, were associated with diabetic neuropathy which is a type of nerve damage in patients with diabetes [43,44]. Consistent with previous studies, our results show that the impairment of glycolysis contributes to the cell death of auditory cells. Our results suggest that the regulation of glycolysis might be critical for the viability of auditory cells for normal functions of hearing.

Alcohol exposure has been linked to hearing impairment [45,46,47,48,49]. A recent study suggests that prenatal alcohol exposure is significantly associated with suspected hearing impairment during early childhood [45]. The results showed that children with prenatal alcohol exposure had a higher risk of suspected hearing impairment compared to the unexposed control [45]. Additionally, acute alcohol exposure was associated with reversible changes of hearing, including temporary worsening of auditory thresholds, poorer speech discrimination, elevation of the acoustic reflex threshold, and impaired processing of sounds [44,45]. Chronic alcohol exposure was also associated with irreversible hearing loss and the prevalence of hearing loss [46,47]. Chronic alcohol exposure also induces the organ injury and cell death. Previous studies showed that chronic high alcohol exposure promotes apoptosis in cardiac cells [48,49]. Also, chronic alcohol exposure caused apoptotic cell death and impaired autophagy in the liver [50,51]. Consistent with previous studies, our results suggest that alcohol exposure induces cytotoxicity via apoptosis. During the alcohol exposure, the cellular signaling pathway, which regulates cell proliferation and survival, is inhibited. Previous reports showed that alcohol suppressed the PI3K/AKT signaling pathway in liver cells [52,53]. However, it is still unclear to determine the level of inner ear alcohol content by alcohol exposure in BAC of alcohol content. Also, the level of inner ear alcohol content is not only affected by BAC of alcohol content but by other factors such as an individual’s heathy condition and genetics. The inner ear alcohol content could be higher or lower than BAC of 0.04% as the moderate alcohol exposure. In our study, we used two concentrations of alcohol exposure with 0.01% (low) and 0.05% (high) in our system as low inner ear alcohol content and high inner ear alcohol content. Consistent with previous studies, our results suggest that alcohol exposure suppresses the PI3K/AKT signaling pathway via the reduction of AKT phosphorylation.

In an upstream target of PI3K/AKT signaling pathway, EGFR has been identified as an important molecule which is critical for cell survival, proliferation, migration, and differentiation [54,55]. Since EGFR signaling is linked to the activation of glycolysis [15,16,17], the inhibition of EGFR increased cell death via apoptosis in cancer such as lung cancer [56,57]. In our results, EGFR signaling was reduced by alcohol exposure. Also, inhibition of EGFR suppressed HK1-dependent glycolysis. These results suggest that the regulation of EGFR-dependent glycolysis could be a critical metabolic pathway for cell survival in auditory cells such as inner or outer hair cells. Since our results showed the role of alcohol exposure in HEI-OC1 cells, there is the limitation to demonstrate the changes of mature inner ear cells by alcohol exposure in vivo. We believe the additional in vivo experiment using a mouse model could be helpful to determine the role of alcohol exposure in mature inner ear cells. Therefore, further work will be needed to overcome our limitation.

Although the pathways that EGFR regulates PI3K-AKT pathway and glucose metabolism were addressed by previous studies [15,16,17,40,41,56,57], our new findings demonstrate that alcohol exposure suppresses the glycolysis via EGFR-HK1 pathway in HEI-OC1 cells. Our new findings showed that alcohol exposure suppresses the expression of EGFR. Moreover, our new findings showed that alcohol exposure suppresses the expression of HK1 via the inhibition of EGFR-AKT pathway. Furthermore, our new findings showed alcohol exposure suppresses the glycolysis activity by downregulation of HK1 using ECAR measurement (Figure 5). Since there is no evidence or report to demonstrate how alcohol exposure could affect the glycolysis during glucose metabolism in HEI-OC1 cells, our new findings might be important for the understanding the novel role of glycolysis for the survival of inner ear cells under alcohol exposure related hearing loss. 

In conclusion, our results demonstrate that impaired glycolysis promotes alcohol exposure-induced apoptosis in HEI-OC1 cells via inhibition of EGFR signaling.

## 4. Materials and Methods

### 4.1. Reagents and Antibodies

Ethanol (E7023) and 2DG (D8375) were from Sigma-Aldrich (St Louis, MO, USA). Erlotinib HCl (OSI-744) (S1023) was from Selleckchem (Houston, TX, USA). The following antibodies were used: monoclonal rabbit anti-HK1 (1:1000) (2024, Cell Signaling Technology (Danvers, MA, USA)), monoclonal rabbit anti-EGFR (1:1000) (ab52894, Abcam, (Cambridge, MA, USA)), polyclonal rabbit anti-Caspase-3 (1:1000) (#9662, Cell signaling Technology (Danvers, MA, USA)), monoclonal rabbit anti-Akt (1:1000) (#9272, Cell signaling Technology (Danvers, MA, USA)), monoclonal rabbit anti-phospho-Akt (Ser473) (#9271, Cell signaling Technology (Danvers, MA, USA)) and monoclonal mouse anti-β-actin (1:5000) (A5316, Sigma-Aldrich (St Louis, MO, USA)).

### 4.2. Cell Culture

HEI-OC1 cells, an auditory cell line derived from the mouse organ of Corti, which possess hair cell-like properties, express several specific molecular markers of hair cells, including Math1 and Myosin VII α [34]. HEI-OC1 cells were incubated in high glucose Dulbecco’s modified Eagle’s medium (DMEM) (WELGENE, KOREA) containing 10% (*v*/*v*) heat-inactivated FBS, 100 units/mL penicillin, 100 mg/mL streptomycin. For alcohol treatment, HE-OC1 cells (2 × 10^5^ cells in 6-well cell culture plate) were incubated with ethanol (0.05% or 0.01% (*v*/*v*), 4 or 24 h). The cell supernatants and cell lysates were collected and analyzed for the LDH assay and levels of HK1 protein by immunoblot. 

### 4.3. 3D Images

HEI-OC1 cells (2 × 10^5^ cells in 6-well cell culture plate) were incubated with ethanol (0.05% (*v*/*v*), 4 h). Further, 3D images were analyzed by 3D Cell Explorer (NANOLIVE, Ecublens, Switzerland). The images were representative images from total 100 cells in ten individual images per group.

### 4.4. Cell Cytotoxicity Assay

Cell cytotoxicity was measured from culture medium of HEI-OC1 cells by LDH-Cytotoxicity Colorimetric Assay Kit II (#K313-500, BioVision (Milpitas, CA, USA)) following the manufacturer’s instructions.

### 4.5. Apoptosis/Necrosis Assay

Apoptosis/Necrosis of cells was measured by GFP CERTIFIED^®^ Apoptosis/Necrosis detection kit for microscopy and flow cytometry (ENZ-51002-100, Enzo Life Sciences (Farmingdale, NY, USA)) according to the manufacturer’s instructions.

### 4.6. Glycolysis Activity Assay

For the glycolytic function assay, HEI-OC1 cells (5 × 10^4^ cells/well) were plated on XF96 cell culture microplates (101085-004, Agilent Technologies, Inc. (Santa Clara, CA, USA)). The ECAR, which is a parameter of glycolytic flux and activity, was measured by a Seahorse XF96e bioanalyzer using the XF Glycolysis Stress Test Kit (102194-100, Agilent Technologies, Inc. (Santa Clara, CA, USA)) according to the manufacturer’s instructions. The ECAR levels were monitored and measured in cells that were treated with glucose (10 mM), oligomycin (2 μM) and 2-deoxyglucose (2DG) (10 mM).

### 4.7. Immunoblot Analysis

Cells were harvested and lysed in NP40 Cell Lysis Buffer (FNN0021, Invitrogen Life Technologies (Grand Island, NY, USA)). Lysates were centrifuged at 15,300× *g* for 10 min at 4 °C, and the supernatants were obtained. The protein concentrations of the supernatants were determined by using the Bradford assay (500-0006, Bio-Rad Laboratories (Hercules, CA, USA)). Proteins were electrophoresed on NuPAGE 4%−12% Bis-Tris gels (Invitrogen (Waltham, MA, USA)) and transferred to Protran nitrocellulose membranes (10600001, GE Healthcare Life science (Pittsburgh, PA, USA)). Membranes were blocked in 5% (*w*/*v*) bovine serum albumin (BSA) (9048-46-8, Santa Cruz Biotechnology (Dallas, TX, USA)) in TBS-T (TBS (170-6435, Bio-Rad Laboratories (Hercules, CA, USA)) and 1% (*v*/*v*) Tween-20 (170-6531, Bio-Rad Laboratories (Hercules, CA, USA)) for 30 min at 25 °C. Membranes were incubated with primary antibody diluted in 1% (*w*/*v*) BSA in TBS-T for 16 h at 4 °C and then with the horseradish peroxidase (HRP)-conjugated secondary antibody (goat anti–rabbit IgG–HRP (SC-2004) (1:2500), goat anti–mouse IgG–HRP (SC-2005) (1:2500), goat anti–rabbit IgG–HRP (SC-2004) (1:2500) from Santa Cruz Biotechnology (Dallas, TX, USA)) diluted in TBS-T for 30 min at room temperature. The immunoreactive bands were detected by the SuperSignal West Pico Chemiluminescent Substrate (34078, Thermo Scientific (Waltham, MA, USA)).

### 4.8. Statistical Analysis

All data are mean ± SEM, combined from three independent experiments. All statistical tests were analyzed by Student’s two-tailed t-test for comparison of two groups, and analysis of variance (ANOVA) (with post hoc comparisons using Dunnett’s test), using a statistical software package (GraphPad Prism version 4.0 (San Diego, CA, USA)) for comparison of multiple groups. Further *p*-values of less than 0.05 were considered statistically significant.

## 5. Conclusions

Our results demonstrate that impaired glycolysis promotes alcohol exposure-induced apoptosis in HEI-OC1 cells via inhibition of EGFR signaling.

## Figures and Tables

**Figure 1 ijms-21-00476-f001:**
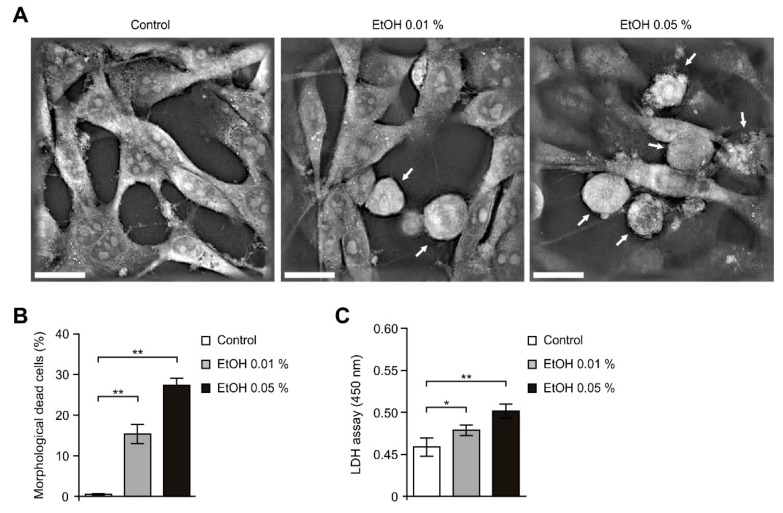
Alcohol exposure induced cytotoxicity in HEI-OC1 cells. (**A**) Representative 3D images of cell death in HEI-OC1 cells treated with ethanol (EtOH, 0.01% or 0.05%) or vehicle (Control) for 4 h. White arrows indicate cells which have the morphological features of cell death. Scale bars, 20 μm. (**B**) Quantification of the morphological dead cells in HEI-OC1 cells treated with ethanol (EtOH, 0.01% or 0.05%) or vehicle (Control) for 4 h (The percent of morphological dead cells in total 100 cells in 10 individual images per group). (**C**) Cytotoxicity assay of HEI-OC1 cells treated with ethanol (EtOH, 0.01% 0.05%) or vehicle (Control) for 4 h using lactate dehydrogenase (LDH) levels in the culture medium. Data are representative of three independent experiments, and each was done in triplicate. Data are mean ± SEM. * *p* < 0.05, ** *p* < 0.01 using the two-tailed Student’s *t*-test.

**Figure 2 ijms-21-00476-f002:**
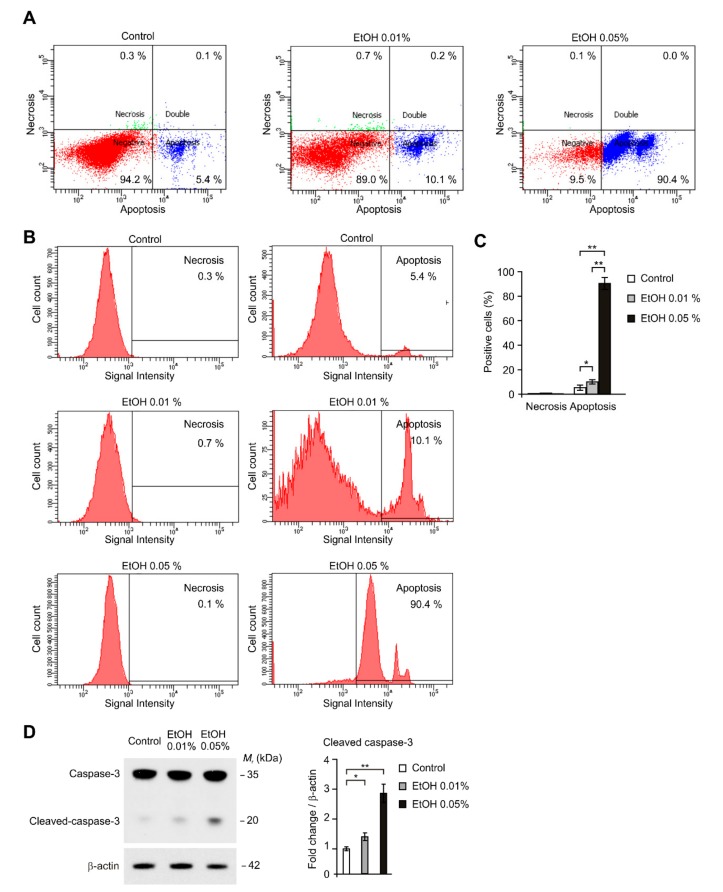
Alcohol exposure induced apoptosis in HEI-OC1 cells. (**A**) Flow cytometry analysis for apoptosis/necrosis detection of HEI-OC1 cells treated with ethanol (EtOH, 0.01% or 0.05%) or vehicle (Control) for 4 h and then stained with Annexin V for necrosis and red-emitting dye 7-AAD for apoptosis. (**B**) The cell counts of apoptosis/necrosis positive cells by flow cytometry analysis in HEI-OC1 cells treated with ethanol (EtOH, 0.01% or 0.05%) or vehicle (Control) for 4h and then stained with Annexin V for necrosis and red-emitting dye 7-AAD for apoptosis. (**C**) Quantification of flow cytometry analysis for apoptosis/necrosis detection of HEI-OC1 cells treated with ethanol (EtOH, 0.01% or 0.05%) or vehicle (Control) for 4h and then stained with Annexin V for necrosis and red-emitting dye 7-AAD for apoptosis. (**D**) Representative immunoblot analysis for cleaved caspase-3 (**left**) and quantification for cleaved caspase-3 protein levels (**right**) from HEI-OC1 cells treated with ethanol (EtOH, 0.01% or 0.05) or vehicle (Control) for 4 h. For immunoblots, β-actin was used as loading control. Data are representative of three independent experiments, and each was done in triplicate. Data are mean ± SEM. * *p* < 0.05, ** *p* < 0.01 using the two-tailed Student’s *t*-test.

**Figure 3 ijms-21-00476-f003:**
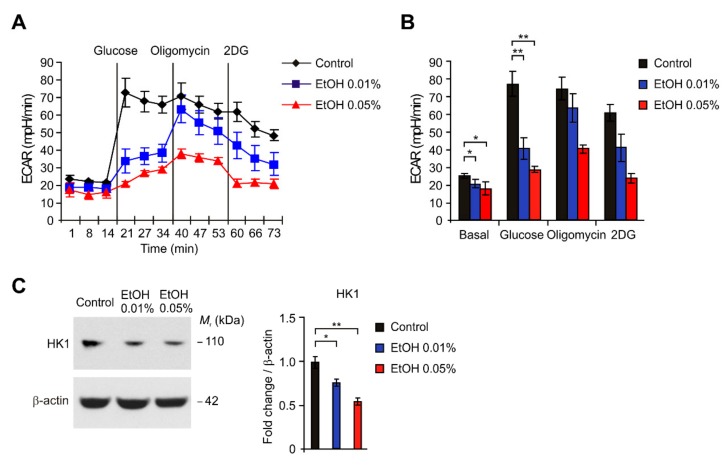
HK1-dependent glycolysis is suppressed by alcohol exposure in HEI-OC1 cells. (**A**) The levels of extracellular acidification rate (ECAR) for glycolysis of glucose and (**B**) quantification of ECAR levels from HEI-OC1 cells treated with ethanol (EtOH, 0.01% or 0.05%) or vehicle (Control) for 4 h. Data are representative of three independent experiments. Data are mean ± SEM. ** *p* < 0.01, * *p* < 0.05 using the two-tailed Student’s t-test. (**C**) Representative immunoblot analysis for HK1 (**left**) and quantification for HK1 protein levels (**right**) from HEI-OC1 cells treated with ethanol (EtOH, 0.01% or 0.05%) or vehicle (Control) for 4h. For immunoblots, β-actin was used as loading control. Data are representative of three independent experiments. Data are mean ± SEM. ** *p* < 0.01, * *p* < 0.05 using the two-tailed Student’s *t*-test.

**Figure 4 ijms-21-00476-f004:**
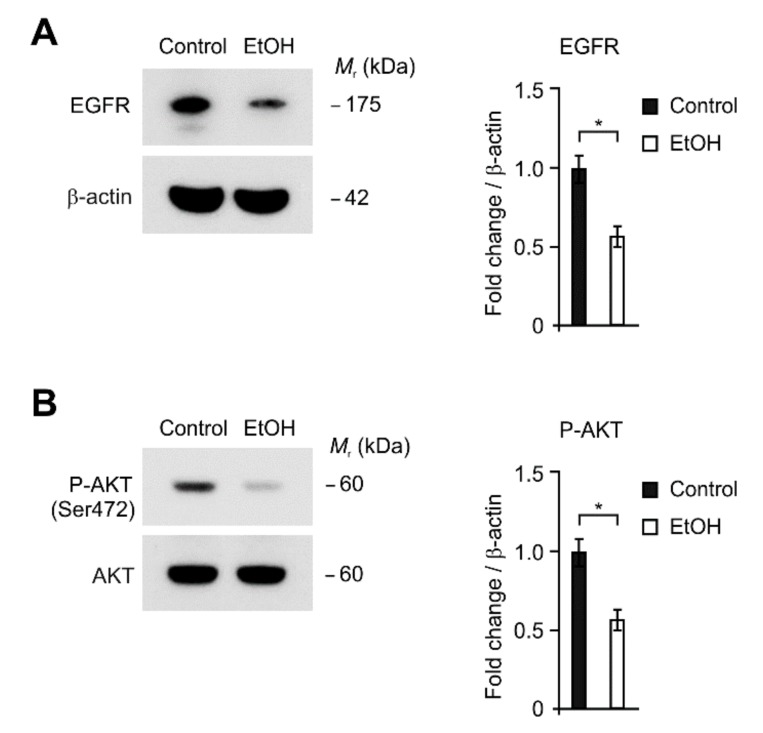
The levels of EGFR and AKT phosphorylation were reduced by alcohol exposure in HEI-OC1 cells. (**A**) Representative immunoblot analysis for HK1 (**left**) and quantification for HK1 protein levels (**right**) from HEI-OC1 cells treated with ethanol (EtOH, 0.05%) or vehicle (Control) for 4 h. For immunoblots, β-actin was used as loading control. Data are representative of three independent experiments. (**B**) Representative immunoblot analysis for AKT phosphorylation (**left**) and quantification for AKT phosphorylation levels (**right**) from HEI-OC1 cells treated with ethanol (EtOH, 0.05%) or vehicle (Control) for 4 h. For immunoblots, total AKT was used as loading control. Data are representative of three independent experiments. Data are mean ± SEM. * *p* < 0.05 using the two-tailed Student’s *t*-test.

**Figure 5 ijms-21-00476-f005:**
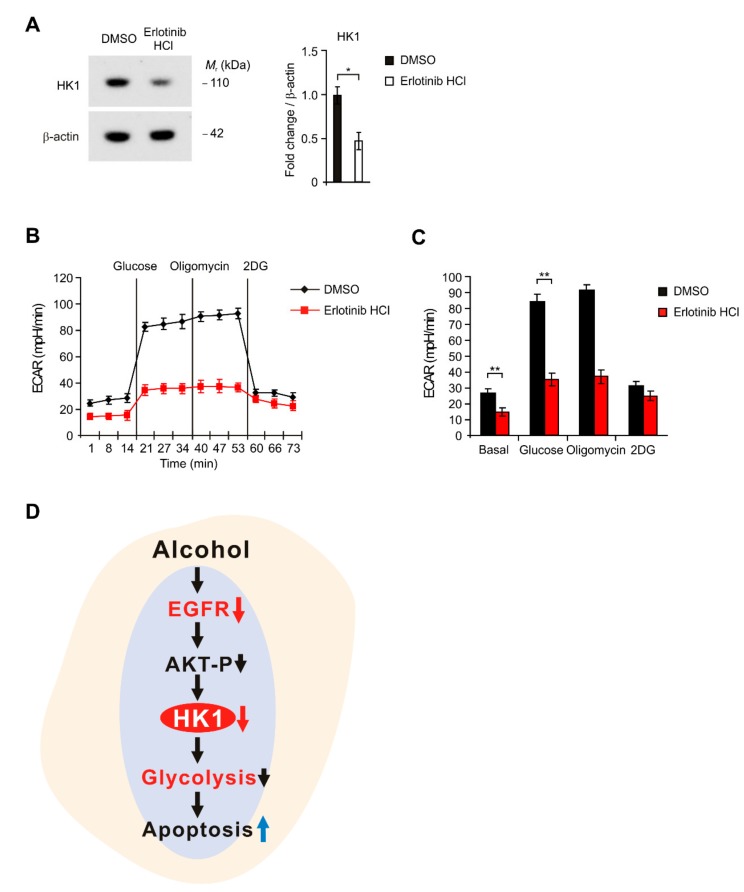
Inhibition of EGFR suppresses HK1-dependent glycolysis in HEI-OC1 cells. (**A**) Representative immunoblot analysis for HK1 (**left**) and quantification for HK1 protein levels (**right**) from HEI-OC1 cells treated with Erlotinib HCl (Erlotinib, 5 μM) or control (DMSO) for 4h. For immunoblots, β-actin was used as loading control. Data are representative of three independent experiments. Data are mean ± SEM. * *p* < 0.05 using the two-tailed Student’s *t*-test. (**B**) The levels of extracellular acidification rate (ECAR) for glycolysis of glucose and (**C**) quantification of ECAR levels from HEI-OC1 cells treated with Erlotinib HCl (Erlotinib, 5 μM) or control (DMSO) for 4 h. Data are representative of three independent experiments. Data are representative of three independent experiments. Data are mean ± SEM. ** *p* < 0.01 using the two-tailed Student’s t-test. (**D**) The schematic diagram to summary of our new findings.

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
