# Peer review of "Impaired Glycolysis Promotes Alcohol Exposure-Induced Apoptosis in HEI-OC1 Cells via Inhibition of EGFR Signaling"

_ijms, 2020, doi:10.3390/ijms21020476_

Round 1

Reviewer 1 Report

The authors now state the novelty and the limitation of this study cleariy. This version of paper can be accepted for publication by IJMS.

Reviewer 2 Report

The authors have satisfactorily completed all this reviewer’s suggested additions and changes. The new data greatly adds to the findings and overall impact of the study. I appreciate and applaud the authors for acting on every one of my suggestions and comments. Thus, the manuscript is suitable in its present revised form for publication.

This manuscript is a resubmission of an earlier submission. The following is a list of the peer review reports and author responses from that submission.

Round 1

Reviewer 1 Report

EGFR signaling pathwayの一つにPI3K-AKTがある(15)。

EGFRがmetabolic pathwayに重要(34)。

The authors’ findings are:

1) Alcohol exposure inhibits EGFR and AKT phosphorylation

2) Inhibition of EGFR-induced glycolysis (metabolic dysfunction), EGFR dependent suppression of HK1 level

3) Apoptosis of HEI-OC1 cells

Only some part of this pathway is supported by previous reports.  I found only two citations, “PI3K-AKT is one of the EGFR signaling pathway (15)” and “EGFR-dependent signaling pathway is critical for glucose metabolism (34)”.  If these pathway, such as “Alcohol exposure inhibits EGFR and AKT phosphorylation”,  “Inhibition of EGFR-induced glycolysis cause apoptosis”, etc, were reported in other cell or organ,  authors need to cite and discuss more on the text.  If these findings are novel even in other cells, authors need to emphasize the novelty, with reason why they focused on these pathways.

HEI-OC1 cells were derived from organ of Corti, but their nature is different from inner ear cells.  They do not have stereocilia as hair cells, they do not have EP generation activity as stria vascularis cells, they do not have synaptic activity as spiral ganglion cells.  It is in doubt that HEI-OC1 cells can be used as the representative for inner ear cells.  Authors need to show the rationale.

The authors need to discuss the alcohol delivery from blood into inner ear for the rationale of 0.05% alcohol.  Is it already proved that 0.12% blood alcohol make inner ear concentration 0.05%?

Author Response

Response to Cells Reviewer 1 Comments

The authors’ findings are:

1) Alcohol exposure inhibits EGFR and AKT phosphorylation

2) Inhibition of EGFR-induced glycolysis (metabolic dysfunction), EGFR dependent suppression of HK1 level

3) Apoptosis of HEI-OC1 cells

Only some part of this pathway is supported by previous reports.  I found only two citations, “PI3K-AKT is one of the EGFR signaling pathway (15)” and “EGFR-dependent signaling pathway is critical for glucose metabolism (34)”.  If these pathway, such as “Alcohol exposure inhibits EGFR and AKT phosphorylation”,  “Inhibition of EGFR-induced glycolysis cause apoptosis”, etc, were reported in other cell or organ,  authors need to cite and discuss more on the text.  If these findings are novel even in other cells, authors need to emphasize the novelty, with reason why they focused on these pathways.

Response 1 :

According to the Reviewer’s request, we provided additional citation for previous report which showed that PI3K-AKT is one of the EGFR signaling pathway in the Introduction section. And, we provided additional citation for previous report which showed that EGFR-dependent signaling pathway is critical for glucose metabolism in the Introduction section. Also, we provided additional citation and description for previous report which showed that alcohol inhibits PI3K-AKT pathway in the Discussion section. We provided additional citation and description for previous report which showed that Inhibition of EGFR-induced glycolysis cause apoptosis in the Discussion section.

The following text has been added to the Introduction, Results and Discussion section:

Introduction, page 1 line 39.  “Additionally, other studies have implicated that the downstream signaling molecules of the epidermal growth factor receptor (EGFR) signaling pathway such as phosphoinositide 3-kinase (PI3K)-AKT could contribute to cell proliferation and survival [15-17].”

Results, page 6 line 148.  “Since EGFR-dependent signaling pathway is critical for glucose metabolism [38-40],”

Discussion, page 8 line 220.  “During the alcohol exposure, the cellular signaling pathway, which regulates cell proliferation and survival, is inhibited. Previous reports showed that alcohol suppressed the PI3K/AKT signaling pathway in liver cells [53,54]. Consistent with previous studies, our results suggest that alcohol exposure suppresses the PI3K/AKT signaling pathway via the reduction of AKT phosphorylation.

In an upstream target of PI3K/AKT signaling pathway, EGFR has been identified as an important molecule which is critical for cell survival, proliferation, migration, and differentiation [55,56]. Since EGFR signaling is linked to the activation of glycolysis [15-17], the inhibition of EGFR increased cell death via apoptosis in cancer such as lung cancer [57,58]. ”

HEI-OC1 cells were derived from organ of Corti, but their nature is different from inner ear cells.  They do not have stereocilia as hair cells, they do not have EP generation activity as stria vascularis cells, they do not have synaptic activity as spiral ganglion cells.  It is in doubt that HEI-OC1 cells can be used as the representative for inner ear cells.  Authors need to show the rationale.

Response 2 :

According to the Reviewer’s request, we provided the rationale for the utilization of HEI-OC1 cells in our study in the Introduction section. HEI-OC1 is one of the few mouse auditory cell lines available for research purposes as a cochlear cell line as in vitro system. We described the explanation of HEI-OC1 in the Materials and Methods section.

The following text has been added to the Introduction and Materials and Methods section:

Introduction, page 2 line 54.  “In the current study, we demonstrate that the impaired glycolysis promotes alcohol exposure-induced apoptosis in HEI-OC1 cells, which is one of the few mouse auditory cells [34-36]”

Materials and Methods, page 8 line 244.  “HEI-OC1 cells, an auditory cell line derived from the mouse organ of Corti, which possess hair cell-like properties, express several specific molecular markers of hair cells, including Math1 and Myosin VII α [34]. “

The authors need to discuss the alcohol delivery from blood into inner ear for the rationale of 0.05% alcohol.  Is it already proved that 0.12% blood alcohol make inner ear concentration 0.05%?

Response 3 :

According to the Reviewer’s request, we provided the more appropriate rationale for 0.05% alcohol in our study via recent paper. Butts M et al showed that a moderate dose of ethanol, defined as the amount in two alcoholic beverages per day for men and one for women, equivalent to a blood alcohol content (BAC) of 0.04%, inhibited the function of intestinal epithelial cells.

The following text has been added to the Results section:

Results, page 2 line 68.  “Since moderate alcohol exposure, defined as the amount in two alcoholic beverages per day for men and one for women, equivalent to a blood alcohol content (BAC) of 0.04%, inhibited the function of intestinal epithelial cells [37],”

Reviewer 2 Report

Kang et al. present a study where they propose that ethanol specifically activates apoptosis in the auditory cell line, HEI-OC1. Further, they observe that ethanol leads to decreased glycolysis via decreased expression of hexokinase 1 and the mechanism further involves inhibition of the epidermal growth factor receptor (EGFR)/AKt pathway.  They postulate that ethanol is a risk factor or even may be a toxicological cause of auditory loss.

The result is a significant study since this reviewer does not recall studies showing the activation of apoptosis by ethanol in auditory cells.  Further, the pathogenesis of such toxicity involving glycolysis and EGFR is intriguing. The main critique is the lack of consistent analysis of ethanol doses in each experiment, where doses of 0.01%, 0.05%, or 0.1% were utilized individually or in dual-combination throughout the study, but not collectively in each study (that is, each experiment did not use all three doses). However, just two doses, 0.01% and 0.05%, would satisfy this reviewer. Overall, the manuscript is well-written, with only a few typos/errors, but also some clarifications are needed.

Therefore, this manuscript is suitable for publication in IJMS with minor revisions.

Comments:

1.  Formatting note: it is usual practice to define acronyms the first time they are listed (i.e. in the abstract).

2.  Fig. 1: a dose-response of ethanol’s effects on the cells would more clearly illustrate the toxicological outcome of such treatments

3.  Fig. 2: shows fantastic data, where clearly, only apoptosis is activated. Again, it would be great to see the effect of 0.01% here. Also, an immunoblot for activated caspases would benefit the study.

4.  Fig. 3: the sequential dosing with glucose, oligomycin, and 2-DG is a bit perplexing, but this reviewer is deducing that such a sequential treatment is designed to analyze glycolysis alone since mitochondrial oxidative phosphorylation is inhibited? Is this a common treatment when performing biochemical studies? Can the authors provide a reference?

5.  Fig. 3A: interesting data, where 0.05% ethanol, or almost 1/2 of the legal limit for driving, almost completely shuts down glycolysis. Because of this experiment, this reviewer would only recommend two doses, 0.01% and 0.05% for each study, since 0.1% would be a bit overkill for these cells.

6.  Fig. 5:  5 µM erlotinib is a bit high, as 1 µM has been shown to effectively inhibit EGFR. Why was this dose utilized?

7.  Mechanistic/technical questions:

-is culturing the cells in high glucose medium problematic for the studies presented?

-are the effects of ethanol due to ethanol itself or its metabolites? Do HEI-OC1 cells contain alcohol/aldehyde dehydrogenase?

-treatment of ethanol ranges from minutes to hours in the studies. What type of exposure to ethanol would individuals need to have in order to affect auditory cells, constant or intermittent? Short or long term?

Author Response

Response to Cells Reviewer 2 Comments

Kang et al. present a study where they propose that ethanol specifically activates apoptosis in the auditory cell line, HEI-OC1. Further, they observe that ethanol leads to decreased glycolysis via decreased expression of hexokinase 1 and the mechanism further involves inhibition of the epidermal growth factor receptor (EGFR)/AKt pathway.  They postulate that ethanol is a risk factor or even may be a toxicological cause of auditory loss.

The result is a significant study since this reviewer does not recall studies showing the activation of apoptosis by ethanol in auditory cells.  Further, the pathogenesis of such toxicity involving glycolysis and EGFR is intriguing. The main critique is the lack of consistent analysis of ethanol doses in each experiment, where doses of 0.01%, 0.05%, or 0.1% were utilized individually or in dual-combination throughout the study, but not collectively in each study (that is, each experiment did not use all three doses). However, just two doses, 0.01% and 0.05%, would satisfy this reviewer. Overall, the manuscript is well-written, with only a few typos/errors, but also some clarifications are needed.

Therefore, this manuscript is suitable for publication in IJMS with minor revisions.

Comments:

Formatting note: it is usual practice to define acronyms the first time they are listed (i.e. in the abstract).

Response 1 :

According to the Reviewer’s request, we provided original information for acronyms in the Abstract section.

The following text has been added to the Abstract and Introduction section:

Abstract, page 1 line 19.  “epidermal growth factor receptor (EGFR)-induced glycolysis is a critical mechanism for alcohol exposure-induced apoptosis in HEI-OC1 cells. The cytotoxicity via apoptosis was significantly increased by alcohol exposure in HEI-OC1 cells. The glycolytic activity and the levels of hexokinase 1 (HK1) ”

Introduction, page 1 line 39.  “Additionally, other studies have implicated that the downstream signaling molecules of the epidermal growth factor receptor (EGFR) signaling pathway such as phosphoinositide 3-kinase (PI3K)-AKT could contribute to cell proliferation and survival [15-17].”

Fig. 1: a dose-response of ethanol’s effects on the cells would more clearly illustrate the toxicological outcome of such treatments

Response 2 :

According to the Reviewer’s request, we provided the additional results for ethanol’s 0.01 % dose in Fig.1A-C. We presented the results for two doses of ethanol (0.01 % and 0.05 %) as a dose-response of ethanol’s effects.

The following text has been added to the Results section:

Results, page 2 line 70.  “we used alcohol exposure of 0.01 % and 0.05 % in HEI-OC1 cells. Alcohol exposure increased the morphological features of cell death such as shrinkage, blebbing and losing contact with neighboring cells relative to control in dose-dependent manner (Figure 1A). Moreover, alcohol exposure significantly reduced the number of cells which were exhibited normal spindle shape in dose-dependent manner (Figure 1B). Next, we analyzed the changes of cell viability by alcohol exposure in HEI-OC1 cells. Cell cytotoxicity was measured by lactate dehydrogenase (LDH) released into the culture media from damaged or dead cells (Figure 1C). Notably, alcohol exposure significantly increased the levels of LDH relative to control in dose-dependent manner (Figure 1C). These results suggest that alcohol exposure induced cytotoxicity in HEI-OC1 cells.”

Fig. 2: shows fantastic data, where clearly, only apoptosis is activated. Again, it would be great to see the effect of 0.01% here. Also, an immunoblot for activated caspases would benefit the study.

Response 3 :

According to the Reviewer’s request, we provided the additional results for ethanol’s 0.01 % dose in Fig. 2A-D. We presented the results for two doses of ethanol (0.01 % and 0.05 %) as a dose-response of ethanol’s effects. Moreover, we provided the additional result for activated caspase-3 in HEI-OC1 cells treated ethanol (0.01 % and 0.05 %) by an immunoblot analysis.

The following text has been added to the Results section:

Results, page 3 line 93.  “Alcohol exposure significantly increased the apoptotic death cells (Annexin V-positive/7-AAD-negative) compared to control in dose-dependent manner (Figure 2A). The number of cells staining for apoptosis (Annexin V-positive/7-AAD-negative) was significantly increased from 5.4 % in control to 10.1 % (1.87 fold) after 0.01 % alcohol treatment or 90.4 % (16.7 fold) after 0.05 % alcohol treatment (Figure 2B, C). In contrast, the number of cells staining for necrosis (Annexin V-negative/7-AAD-positive) was comparable between control and alcohol treatment (Figure 2B, C). Consistently, the levels of cleaved caspase-3, which is activated in the apoptotic cell, were significantly increased by alcohol exposure in dose-dependent manner (Figure 2D).”

Fig. 3: the sequential dosing with glucose, oligomycin, and 2-DG is a bit perplexing, but this reviewer is deducing that such a sequential treatment is designed to analyze glycolysis alone since mitochondrial oxidative phosphorylation is inhibited? Is this a common treatment when performing biochemical studies? Can the authors provide a reference?

Response 4 :

According to the Reviewer’s request, we provided references for the measurement of ECAR as using glucose, oligomycin, and 2-DG. It is a common strategy to understand the entire glucose metabolism such as the initiation step by glucose substrate, the activation step by inhibition of mitochondrial respiration via oligomycin and the inhibition step by glycolysis inhibitor via 2DG as well as previous papers.

The following text has been added to the Results section:

Results, page 5 line 121.  “We measured the extracellular acidification rate (ECAR) as the parameter of glycolysis activity by the quantification of lactate production [38,39]. The glycolytic activity was measured by the sequential addition of glucose, which is the substrate of glycolysis, oligomycin, a selective inhibitor for mitochondrial respiration, and 2-deoxyglucose (2-DG), a specific inhibitor of glycolysis [38,39].”

Fig. 3A: interesting data, where 0.05% ethanol, or almost 1/2 of the legal limit for driving, almost completely shuts down glycolysis. Because of this experiment, this reviewer would only recommend two doses, 0.01% and 0.05% for each study, since 0.1% would be a bit overkill for these cells.

Response 5 :

According to the Reviewer’s request, we provided the results from two doses, 0.01 % and 0.05 % ethanol treatment in Fig. 3A-C.

The following text has been added to the Results section:

Results, page 5 line 132.  “Notably, the levels of HK1 protein were significantly suppressed by alcohol exposure of 0.01 % and 0.05 % relative to control (Figure 3C).”

Fig. 5: 5 µM erlotinib is a bit high, as 1 µM has been shown to effectively inhibit EGFR. Why was this dose utilized?

Response 6 : According to the Reviewer’s request, we provided the additional results with 1 µM erlotinib in supplementary figure 1. Since 5 µM erlotinib treatment resulted in the higher reduction of HK1 protein levels than 1 µM erlotinib treatment, we presented the results with 5 µM erlotinib treatment in original manuscript.

The following text has been added to the Results section:

Results, page 7 line 177.  “Moreover, the low dose treatment of Erlotinib HCl (1 µM) also suppressed the levels of HK1 and ECAR compared to control (Figure S1).”

Mechanistic/technical questions:

-is culturing the cells in high glucose medium problematic for the studies presented?

Response 7-1 :

It was fine for culturing cells.

-are the effects of ethanol due to ethanol itself or its metabolites? Do HEI-OC1 cells contain alcohol/aldehyde dehydrogenase?

Response 7-2 :

Based on previous reports, we believe that our results are the effects of ethanol itself. However, it will be an interesting study to investigate the role of ethanol-derived metabolites in further study.

-treatment of ethanol ranges from minutes to hours in the studies. What type of exposure to ethanol would individuals need to have in order to affect auditory cells, constant or intermittent? Short or long term?

Response 7-3 :

We believe that short term exposure is required for auditory cells in vitro system to investigate the selective cellular signaling pathway and metabolism. Long term exposure will shut down entire cellular signaling pathway and metabolism.

Round 2

Reviewer 1 Report

The HEI-OC1 cells were derived from inner ear, but its characteristics as mature inner ear cells are in doubt.  It may acquire its specific characteristics during culture to be cell line.  The experiments have to be performed by explant culture.

Although the moderate alcohol exposure is equivalent to BAC of 0.04%, it does not mean the inner ear alcohol content of 0.04%.  The concentration of alcohol the authors used in this study may not correct.

The pathways are already well addressed by previous studies.  The only novelty of this study is, they confirmed these pathways in HEI-OC1 cells.  Because HEI-OC1 cells may not have the inner ear character, and the exposure concentration of alcohol may not be suitable, I need to say that this study mean nothing and can not be published by IJMS.